# The Intermediate Filament Synemin Regulates Non-Homologous End Joining in an ATM-Dependent Manner

**DOI:** 10.3390/cancers12071717

**Published:** 2020-06-28

**Authors:** Sara Sofia Deville, Anne Vehlow, Sarah Förster, Ellen Dickreuter, Kerstin Borgmann, Nils Cordes

**Affiliations:** 1OncoRay—National Center for Radiation Research in Oncology, Faculty of Medicine Carl Gustav Carus, Technische Universität Dresden, 01307 Dresden, Germany; SaraSofia.Deville@uniklinikum-dresden.de (S.S.D.); anne.vehlow@nct-dresden.de (A.V.); Sarah.Foerster@ukbonn.de (S.F.); ellen.dickreuter@gmail.com (E.D.); 2Helmholtz-Zentrum Dresden—Rossendorf (HZDR), Institute of Radiooncology—OncoRay, 01328 Dresden, Germany; 3National Center for Tumor Diseases, Partner Site Dresden: German Cancer Research Center, 69120 Heidelberg, Germany; 4German Cancer Consortium, Partner Site Dresden: German Cancer Research Center, 69120 Heidelberg, Germany; 5Laboratory of Radiobiology and Experimental Radiation Oncology, University Medical Center Hamburg-Eppendorf, 20246 Hamburg, Germany; borgmann@uke.de; 6Department of Radiotherapy and Radiation Oncology, University Hospital Carl Gustav Carus, Technische Universität Dresden, 01307 Dresden, Germany

**Keywords:** synemin, DNA-PKcs, ATM, DNA repair, NHEJ, radiosensitivity, HNSCC

## Abstract

The treatment resistance of cancer cells is a multifaceted process in which DNA repair emerged as a potential therapeutic target. DNA repair is predominantly conducted by nuclear events; yet, how extra-nuclear cues impact the DNA damage response is largely unknown. Here, using a high-throughput RNAi-based screen in three-dimensionally-grown cell cultures of head and neck squamous cell carcinoma (HNSCC), we identified novel focal adhesion proteins controlling DNA repair, including the intermediate filament protein, synemin. We demonstrate that synemin critically regulates the DNA damage response by non-homologous end joining repair. Mechanistically, synemin forms a protein complex with DNA-PKcs through its C-terminal tail domain for determining DNA repair processes upstream of this enzyme in an ATM-dependent manner. Our study discovers a critical function of the intermediate filament protein, synemin in the DNA damage response, fundamentally supporting the concept of cytoarchitectural elements as co-regulators of nuclear events.

## 1. Introduction

DNA double strand breaks (DSB) are the most lethal damages generated by genotoxic agents, such as ionizing radiation (IR) and chemotherapeutics [1], accounting for the therapeutic benefit of current cancer treatment modalities. These complex DNA lesions are repaired by two major cellular mechanisms: Non-homologous end joining (NHEJ) and homologous recombination (HR) [2,3]. Hence, targeting the DNA repair machinery is considered a potentially effective approach for eradicating cancer cells, and several pharmacological inhibitors are being currently tested in clinical trials [4] (www.clinicaltrials.org).

While HR is mostly regarded as an error-free repair process restricted to the S/G2 phase, NHEJ is error-prone and active throughout the mammalian cell cycle [5]. During NHEJ, DSB are recognized by the binding of the Ku70/Ku80 heterodimers to broken DNA ends, followed by the recruitment and activation of the apical DNA repair kinases, DNA protein kinase catalytic subunit (DNA-PKcs) and Ataxia telangiectasia mutated (ATM) [2]. Subsequently, the nucleases Artemis, Aprataxin or APLF are recruited to complete the end processing, together with ligases IV, XRCC4, XLF and PAXX, as well as numerous chromatin and nuclear matrix remodeling enzymes to allow for the maintenance of genomic stability [5].

In addition to our deep understanding of nuclear DSB repair events, recent work has identified several extra-nuclear factors, which fundamentally and context-dependently modify DNA repair via yet to be determined mechanisms. Examples of such key co-regulators are growth factor receptors and integrin cell adhesion molecules, connecting tumor cells to the extracellular matrix (ECM), and its stiffness in juxtamembrane multiprotein complexes, called focal adhesions [6,7,8]. To date, a variety of focal adhesion proteins (FAP), such as β1 integrin, PINCH1, FHL2, focal adhesion kinase, vimentin and caveolin-1 have been demonstrated to significantly confer tumor resistance to genotoxic agents by modulating the repair of DSB upon radiochemotherapy [9,10,11,12,13,14]. In addition, accumulating evidences suggest that cytoplasmic FAP signaling is linked to nuclear repair dynamics through components of the DNA repair machinery, including DNA-PKcs, ATM, BRCA1, RAD51 and c-Abl [15].

To systematically search for novel FAP candidates driving the cancer therapy resistome by linking extracellular cues to intracellular DNA repair decisions, we conducted a high-throughput RNA interference screen combined with X-ray irradiation in more physiologically three-dimensional (3D) ECM based cell cultures of human head and neck squamous cell carcinoma (HNSCC). We identified the large type IV intermediate filament (IF) protein, synemin [16,17,18] as a critical determinant of cellular radioresistance and NHEJ-related DSB repair in HSNCC. We mechanistically demonstrate that synemin governs DNA-PKcs phosphorylation and activity and reveal a fundamental role of synemin in DSB repair by serving as a kinase-anchoring protein for an ATM-dependent interaction with DNA-PKcs. Taken together, our study provides evidence of cytoarchitectural elements, such as intermediate filaments as key co-regulators of nuclear DNA repair.

## 2. Results

### 2.1. Identification of Adhesome-Based Key Regulators of Radiation Sensitivity and DNA Double Strand Break Repair in HNSCC

To systematically investigate the function of individual FAP in HNSCC radioresistance and DSB repair, we performed a high-throughput RNAi-based screen depleting 117 FAP in UTSCC15 HNSCC cells stably expressing EGFP-53BP1 (UTSCC15 EGFP-53BP1), as shown in Appendix A, grown in 3D lrECM (3DHT-RNAi-S), as shown in Figure 1A and Appendix A. While a crucial contribution to cancer cell therapy resistance has been reported by us for several FAP, such as β1 integrin, LIMS1, FAK and FHL2 [19,20,21], here we identified a number of novel candidates. In Figure 1B–D, we plotted the enhancement ratios of non-irradiated and irradiated cells, as well as the gain of EGFP-53BP1 foci, according to the formulas described under the Materials and Methods section. In the absence of irradiation, the colony formation capacity of UTSCC15-EGFP-53BP1 transfectants remained virtually unaffected. Exceptions for significantly induced colony formation were the depletion of, for example, fermitin family member 2 (FERMT2, also known as PLEKHC1) and zyxin and, for reduced colony formation, the depletion of kinesin family member 11 (KIF11), as shown in Figure 1B, Appendix A and in Appendix A. In contrast, colony formation capability was strongly altered with significant reductions upon the knockdown of several FAP, such as vinculin (VCL), growth factor receptor bound protein 7 (GRB7), sorbin and SH3 domain containing 2 (SORBS2) and synemin (SYNM), as shown in Figure 1C, Appendix A and in Appendix A.

In parallel, we quantified the residual EGFP-53BP1 foci as readout for DSB repair capacity in unirradiated and irradiated cell cultures. While basal levels of EGFP-53BP1 foci were unchanged, as shown in Appendix A, the depletion of various FAP induced significant increases in residual EGFP-53BP1 foci, as shown in Figure 1D and Appendix A. Novel identified determinants of DSB repair included the α7, α8, and β8 integrin subunits, as well as kelch-like ECH associated protein 1 (KEAP1), talin 1, LIM domain binding 3 (LDB3), syndecan binding protein (SDCBP), and synemin, as shown in Figure 1D and Appendix A. Plotting the surviving fraction at 6-Gy against gained 53BP1 foci numbers revealed, for some but not all proteins, a functional context between radiation survival and DSB repair, as shown in Figure 1E and Appendix A. Taken together, our results imply that a perturbed function of specific FAP concurrently and significantly alters both colony formation and DSB repair. Moreover, the presented data show that our 3D-HT-RNAi-S is a robust screening platform for the identification of potential regulators of cellular radiation survival and DSB repair.

### 2.2. Synemin Modulates Radiation Sensitivity and DNA Double Strand Break Repair in HNSCC Cells

The novelty of these findings prompted us to focus our further analyses on one of the most promising candidates from our screen, the IF protein synemin. Using the Oncomine database (https://www.oncomine.org) [22], we explored synemin mRNA expression across multiple head and neck cancers and found that synemin was significantly upregulated in head and neck cancers compared to normal tissue, as shown in Figure 2A. In line with these data, synemin was amplified in several squamous cell carcinomas, such as HPV negative HNSCC, lung squamous cell carcinomas (LUSCC) and cervix squamous cell carcinomas (CESCC), as shown in Appendix A. Next, we searched for the predicted protein interactions of synemin with the DNA repair machinery using Cytoscape [23] and identified a potential association of synemin with the DNA repair kinases ATM, ATR and DNA-PKcs, as shown in Appendix A.

Subsequently, we sought to demonstrate that synemin acts as general determinant of both radiation survival and DSB repair. For this, we validated our 3D-HT-RNAi-S results in a panel of 10 3D lrECM grown HNSCC cell lines, which demonstrated synemin expression in both the cytoplasm and nucleus, as shown in Figure 2B,C and Appendix A. Intriguingly, while basal colony formation remained unaffected, as shown in Figure 2D,F, all 10 HNSCC cell lines showed enhanced radiosensitivity upon synemin silencing, relative to the controls, as shown in Figure 2E,F. Accordingly, synemin silencing elicited significantly elevated 53BP1 foci numbers in all tested HNSCC cell lines after X-ray irradiation, relative to the controls, as shown in Figure 2G–I. Confirmatory data for the fundamental role of synemin in both radiation survival and DSB repair were generated in stably mCherry–Synemin-overexpressing SAS cells, as shown in Figure 2J. Synemin overexpression significantly increased colony formation ability and significantly lowered the numbers of residual EGFP-53BP1 foci, relative to the controls, as shown in Figure 2K,L, respectively. Collectively, our results suggest that synemin plays an essential role in cell survival, as well as in DSB repair after genotoxic injury.

### 2.3. Synemin Regulates NHEJ by Influencing DNA-PKcs and ATM Phosphorylation

To unravel the role of synemin in DNA repair in more detail, we conducted DNA repair reporter assays to measure HR and NHEJ activities, as shown in Appendix A. While synemin depletion left HR activity unaffected, NHEJ activity significantly declined by approximately 40%, as shown in Figure 3A,B and Appendix A. In line, synemin overexpression resulted in a five-fold increase in NHEJ activity, as shown in Figure 3C. Subsequently, we examined the phosphorylation and expression of the key NHEJ-associated proteins, DNA-PKcs and ATM, and found that synemin silencing causes a defect in the radiogenic hyperphosphorylation of both DNA-PKcs at S2056 and ATM at S1981, as shown in Figure 3D,E, without affecting their basal expression levels, as shown in Appendix A. In contrast, synemin overexpression enhanced basal and radiogenic levels of phosphorylated S2056-DNA-PKcs, as shown in Figure 3F,G. Next, we analyzed the kinetics of DNA-PKcs S2056, 53BP1 and γH2AX DNA repair foci upon 1-Gy X-ray irradiation in synemin-depleted cells and observed, in line with the reduced phosphorylation of DNA-PKcs, significantly declined DNA-PKcs S2056 foci levels over the 24 h observation period, relative to the controls, as shown in Figure 3H,K and Appendix A. In contrast, 53BP1 and γH2AX foci lacked a significant difference at 30 min post irradiation, but were significantly higher throughout the remaining observation time in the synemin-depleted cells, compared to the controls, as shown in Figure 3I,J and Appendix A. Taken together, our data suggest that synemin critically modulates NHEJ by influencing DNA-PKcs and ATM.

### 2.4. Synemin Influences DSB Repair Responses by Regulating DNA-PKcs

Based on the observed dependency of DNA-PKcs phosphorylation on synemin and without significant differences in cell cycling, as shown in Appendix A, we next investigated the functional relationship between synemin and DNA-PKcs for DSB repair. Single and double depletion of synemin and DNA-PKcs intriguingly revealed similar residual 53BP1 and γH2AX foci numbers, relative to the controls after 6-Gy X-ray exposure, as shown in Figure 4A–C and Appendix A. This suggests that: (i) synemin critically impacts on the functionality of DNA-PKcs in DSB repair; (ii) synemin and DNA-PKcs are components of the same signaling pathway. To underpin this interdependency of synemin and DNA-PKcs, we quantified colony formation after single and double knockdown without and in combination with irradiation. While basal cell survival was non-significantly affected, as shown in Figure 4D, single synemin and DNA-PKcs knockdown significantly enhanced cellular radiosensitivity, relative to the controls, as shown in Figure 4E. Intriguingly, both single DNA-PKcs, as well as double synemin/DNA-PKcs knockdown, induced radiosensitization to an extent superimposable to that observed for single synemin knockdown, as shown in Figure 4E. Thus, our results depict a dependency of the DNA damage response on a functional DNA-PKcs–synemin interaction.

### 2.5. Synemin Interacts with DNA-PKcs in an ATM-Dependent Manner

To investigate a potential interaction between synemin and DNA-PKcs, we conducted immunoprecipitation (IP) assays with endogenous and mCherry–Synemin in unirradiated and X-ray irradiated SAS cells. In unirradiated cells, we found that DNA-PKcs bound to synemin, relative to the IgG and mCherry controls, as shown in Figure 5A,B (A = pulldown of endogenous synemin; B = pulldown of mCherry–Synemin). This interaction, however, partially dispersed in X-ray irradiated cells, as shown in Figure 5A,B. After observing a dependence of DNA-PKcs phosphorylation on synemin, we decided to perform a proximity ligation assay (PLA) between synemin and DNA-PKcs S2056, as shown in Appendix A. Interestingly, we identified an interaction between synemin and the phosphorylated form of DNA-PKcs that was mainly nuclear, as shown in Appendix A.

Based on our finding that not only DNA-PKcs, but also ATM, showed less radiogenic hyperphosphorylation, we hypothesized a putative interplay between synemin, DNA-PKcs and ATM. By means of SKX cells, a squamous cell carcinoma cell line with ATM downregulation by miR-421 overexpression [24], we show loss of DNA-PKcs pulldown in synemin immunoprecipitates from unirradiated and irradiated ATM-depleted SKX cells, as shown in Figure 5C in lanes 6 and 8, relative to the ATM-expressing SAS cells, as shown in Figure 5C in lanes 5 and 7. We further found an increased expression of synemin in SKX cells, suggesting a rescue or response mechanism elicited by ATM dysfunctionality, as shown in Figure 5C. To investigate the functional consequence of the synemin/DNA-PKcs interaction, we explored 53BP1 and DNA-PKcs S2056 DNA repair foci in unirradiated and irradiated mCherry- and mCherry–Synemin-expressing cells pretreated with the ATM inhibitor KU55933. Relative to DMSO-treated mCherry controls, 53BP1 and DNA-PKcs S2056 foci were significantly decreased at 1 h post irradiation and significantly increased at 24 h post 1-Gy irradiation in the KU55933-pretreated mCherry controls, as shown in Figure 5D,E. Importantly, a similar pattern was found in mCherry–Synemin-expressing cells, as shown in Figure 5D,E. Hence, our results show that the general function of synemin in DSB repair and the recruitment of 53BP1 and DNA-PKcs to DSBs is ATM-dependent.

### 2.6. ATM Phosphorylation of the Synemin Tail Modulates DNA Repair

To determine the part of the synemin protein mediating its function in DNA repair, we generated two synemin constructs—one by the deletion of coil–coil linker/tail domains (mCherry–Synemin_Head) and another one by the deletion of head/coil–coil linker domains (mCherry–Synemin_Tail), as shown in Figure 6A and Appendix A. We evaluated 53BP1 and DNA-PKcs S2056 DNA repair foci upon the overexpression of full-length mCherry–Synemin and the deletion variants. We found reduced 53BP1 and increased DNA-PKcs S2056 foci numbers in mCherry–Synemin-overexpressing cells, as shown in Figure 6B,C, respectively. Interestingly, these effects were lost upon the expression of mCherry–Synemin_Head, but not mCherry–Synemin_Tail, as shown in Figure 6B,C, respectively, suggesting the tail of synemin to be key for synemin’s function in DNA repair. Furthermore, the observed overexpression effects of synemin on residual 53BP1 foci were lost post 1-Gy X-ray exposure upon pretreatment with KU55933 indicating a dependency of the function of the synemin tail domain on ATM activity, as shown in Figure 6D.

In order to predict the amino acid residues phosphorylatable by ATM, we used the GSP database and found the S421, S554, S1114 and S1159 amino acid residues at the synemin tail, but none at the head or coil-linker domains of synemin. To characterize the role of these phosphorylation sites at the synemin tail for DNA repair, we generated further synemin constructs containing either tail amino acids 301 to 961 (mCherry–Synemin_301–961) or tail amino acids 962 to 1565 (mCherry–Synemin_962–1565), as shown in Figure 6A and Appendix A. The analysis of 53BP1 repair foci upon 1-Gy X-ray irradiation revealed a significant increase in 53BP1 foci in cells expressing mCherry–Synemin_962–1565, but not mCherry–Synemin_301–961, indicating a potential engagement of S1114 and S1159 in DNA repair, as shown in Figure 6C,D. To further prove the functionality of these serine residues, we introduced point mutations preventing the phosphorylation of either S1114 (mCherry–Synemin_S1114A) or S1159 (mCherry–Synemin_S1159A), as shown in Figure 6A and Appendix A. Additionally, 53BP1 foci quantification upon the overexpression of these constructs revealed that S1114, but not S1159, is essential for the repair of radiogenic DSBs, as shown in Figure 6E and Appendix A.

Subsequently, we explored the phosphorylation kinetics of DNA-PKcs S2056 in mCherry controls, mCherry–Synemin transfectants and mCherry–Synemin S1114A mutants upon 6-Gy X-ray exposure. Intriguingly, the serine 1114 point mutation led to a similar phosphorylation kinetic than mCherry controls, while mCherry–Synemin expression conserved DNA-PKcs phosphorylation over the 24-h observation period, as shown in Figure 6F. These data evidently show that the serine 1114 amino acid residue at the synemin tail is specifically required for its DNA repair function. Interestingly, this serine is surrounded by glutamines (Q) indicating that the amino acid residue sequence from 1113 to 1115 is QSQ, for which a higher specificity for ATM phosphorylation has been documented [25]. Taken together, our study shows that the ATM-dependent interaction of synemin and DNA-PKcs controls NHEJ, cell survival and radiosensitivity. Synemin silencing perturbs DNA-PKcs phosphorylation, reduces cell survival and enhances radiosensitivity by modulating the functionality of NHEJ, as shown in Figure 6G.

## 3. Discussion

The reduction in treatment resistance remains one of the major challenges for improving cancer patient survival. To safeguard genomic stability and survival, normal and transformed cells employ multiple efficient and complex DNA repair mechanisms. While numerous DNA-damaging agents like radio- and chemotherapy are standard of clinical care, the addition of biologicals seems required and advantageous for a more efficacious eradication of the various subpopulations of therapy-sensitive and -resistant malignant cells. Recent work has demonstrated that the two main DNA repair processes (i.e., HR and NHEJ), are more than just nuclear events as they are critically co-regulated by extracellular and cytoplasmic cues [26,27]. Among these factors, numerous transmembrane growth factor and adhesion receptors, as well as cytoplasmic protein kinases and adapter proteins coalescing at focal adhesions, exist. These focal adhesions serve as essential and powerful hubs for pro-survival resistance-mediating and DNA repair-modifying signal transduction [8,11,26,28,29,30,31]. To gain deeper insight into the functions of focal adhesion proteins (FAP) in the therapy resistance of HNSCC cancer cells, we established a 3D high-throughput RNAi-based screen (3DHT-RNAi-S) and identified a previously uncharacterized function of the IF protein, synemin. We show that synemin regulates the auto-phosphorylation of DNA-PKcs (S2056) and the recruitment of DNA-PKcs to DSB for modulating NHEJ and the radiochemosensitization of HNSCC cells, as shown in Figure 6G.

The type IV IF protein synemin, expressed as an alpha and a beta form, is critical for various cell functions and the formation of organs, such as heart and bones [16,32]. Similar to other IF proteins, synemin is overexpressed in different human malignancies, contributing to a more aggressive phenotype [33,34]. In breast cancer, synemin expression is modified by aberrant promoter methylation and correlates with early relapse [35]. In glioblastoma, synemin controls cell proliferation through AKT by antagonizing PP2A [36]. Interestingly, in human hepatocellular carcinomas, a down-regulation of synemin fails to alter the stability of the cytoskeleton, indicating the tissue specificity and multifunctionality of synemin [37]. In line with our findings of a synemin–DNA-PKcs interaction, synemin has been described as an IF protein with an inability to self-assemble into filaments [38]. In turn, these unique biophysical properties of synemin facilitate the assembly of dynamic and content-specific cytoarchitecture- and stress response-related interactomes [39]. The involvement of IF in the cellular stress response is well known for various challenging events, such as tissue repair, heat shock, antimicrobial defense and apoptosis [40,41]. Our observations widen this spectrum to a new facet (i.e., DNA repair upon radiogenic genotoxic stress). Obviously, the cellular response to damaged DNA and the processes required for genome protection and integrity are among a number of commonalities between these stress conditions. It is of utmost interest that the more widely active and error-prone NHEJ repair mechanism seems to be controlled by an IF protein that also forms a protein complex with the essential DNA repair protein, DNA-PKcs. Moreover, our data pinpoint that synemin lies hierarchically upstream of DNA-PKcs. These findings might be contextually linked to coordinated actions between cytoskeletal elements and the DNA found during cell movement and cell division. IF proteins, similar to actin filaments or microtubules, liaise cell membrane and nuclear membrane/envelope through the Linker of the Nucleoskeleton and Cytoskeleton (LINC) complexes [42,43]. This allows evolutionary conserved processes to be conducted, such as transfer of physical forces from the extracellular space onto chromosomes or the cytoplasmic–nuclear transfer of IF proteins. In line with our data, Lottersberger and colleagues noted the dependence of telomeres on the LINC complex, microtubules and 53BP1 [44]. They proposed that increased telomere mobility induces NHEJ and reduces mis-repair through an increased probability for correct ligations.

Another influencing facet is the dependency of the synemin–DNA-PKcs protein complex formation on ATM kinase activity in the absence and presence of radiogenic genotoxic injury. Additionally and irrespective of its kinase activity, DNA-PKcs dispersed from synemin in a radiogenic stress-dependent manner. Using the ATM deficient cell line SKX, we provide evidence that the interrelation between DNA-PKcs and synemin relies on ATM. GPS database computational prediction of phosphorylatable amino acid residues of the synemin tail yielded several potential substrates for ATM, which essentially contribute to DSB repair in an ATM-dependent manner. This result is not very surprising on the basis of ATM´s multiple roles, such as DSB detection and repair, global DNA damage checkpoint activation and the organization of the local and the global chromatin landscape [45]. In addition, it was recently shown that mitochondria exchange and survival and DNA repair signaling is rigorously ATM-dependent under multicellular conditions [46]. In view of our findings, ATM seems localized upstream of a regulatory synemin–DNA-PKcs complex, acting as a superior determinant of DSB repair.

In summary, our results reveal a critical function of the IF protein, synemin in regulating the radiosensitivity of HNSCC cells through the modulation of NHEJ-mediated DSB repair. In an ATM-dependent manner, synemin serves as a scaffold for DNA-PKcs and co-determines DSB repair upon genotoxic injury. Our findings further highlight the complexity of DSB repair by supporting the concept of cytoarchitectural elements as key co-regulators of nuclear events, such as the DNA damage response.

## 4. Materials and Methods

### 4.1. Cell Lines and 3D Cell Culture

HNSCC cell lines (Cal33, FaDu, SAS, UTSCC5, UTSCC8, UTSCC14, UTSCC15, UTSCC45 and XF354fl2) were kindly provided by R. Grenman (Turku University Central Hospital, Turku, Finland). SKX cells were kindly provided by M. Krause (Technische Universität Dresden, Dresden, Germany). Cal33-DRGFP and Cal33-pimEJ5GFP were kindly provided by K. Borgmann (University Medical Center Hamburg-Eppendorf, Hamburg, Germany). Cells were asynchronously grown in Dulbecco’s modified Eagle’s medium containing glutamax-I (from AppliChem, Darmstadt, Germany) supplemented with 10% fetal calf serum and 1% non-essential amino acids (all PAA Laboratories, Toronto, ON, Canada) at 37 °C in a humidified atmosphere containing 8.5% CO_2_. For 3D cell cultures, cells were embedded in 0.5 mg/mL laminin-rich extracellular matrix (lrECM) (Matrigel; BD, Heidelberg, Germany), as described previously [21]. All cell lines were authenticated using STR DNA profiling and tested negative for mycoplasma contamination.

### 4.2. X-Ray Irradiation

Irradiation was performed at room temperature using single doses of 200 kV X-rays (Yxlon Y.TU 320; Yxlon, Hamburg, Germany; dose rate ~1.3 Gy/min at 20 mA), filtered with 0.5 mm Cu, as described previously [47]. Dosimetry for quality assurance was performed using a Duplex dosimeter (PTW Freiburg; Freiburg, Germany) prior to irradiation.

### 4.3. Antibodies

Specific primary and secondary antibodies and their purchasing source are listed in Appendix A.

### 4.4. 3D High-Throughput Screen Using esiRNA (3D HTP-RNAi-S)

We designed a 3D HTP-RNAi-S library of endoribonuclease-prepared siRNAs (esiRNA) [48] for 117 focal adhesion proteins and control esiRNA, as noted in Appendix A, based on [49]. The library was purchased from Eupheria Biotech (Dresden, Germany). UTSCC15 cells stably expressing pEGFP-53BP1-C1 were seeded in 96-well plates and were transfected for 8 h using a solution composed of 18.6 µL Opti-MEM, 1µL esiRNA (concentration, 10 ng/mL) and 0.4 µL oligofectamine (Invitrogen, Karlsruhe, Germany) per well. Subsequently, Opti-MEM plus 10% FCS was added to the cells. After 24 h the cells were re-suspended in 0.5 mg/mL lrECM and seeded in 96-well plates, pre-coated with 1% agarose. The next day, cells were irradiated with 6-Gy X-rays or left untreated. The residual number of foci was evaluated 24 h post irradiation, while colony formation ability was measured after eight days of incubation, as published in [11]. Enhancement ratios (ER) were calculated as follows:$$\mathrm{ER}\;\left(0\;\mathrm{or}\;6\right)=\frac{\sum^{\;}\mathrm{Normalized}\;\mathrm{surviving}\;\mathrm{fraction}\;\mathrm{esiCTRL}}{\sum^{\;}\mathrm{Normalized}\;\mathrm{surviving}\;\mathrm{fraction}\;\mathrm{esiRNA}}$$

The residual foci number was calculated by subtracting the number of the esiCTRL foci to the esiRNA treated cells post 6-Gy X-ray.

### 4.5. esiRNA and siRNA Transfection

Synemin esiRNA (comprised of a heterogeneous pool of siRNA) was purchased from Eupheria Biotech. DNA-PKcs siRNA and Silencer Negative Control siRNA (AM4635) were obtained from Ambion (Darmstadt, Germany). All the esi/siRNA sequences are listed in Appendix A. siRNA transfection was carried out as published in [11]. In brief, 24 h after plating, 26 nM/20 nM esiRNA/siRNA was delivered using 8 µL and 4 µL oligofectamine, respectively, and Opti-MEM (Invitrogen) under serum-free condition for 8 h. Subsequently, Opti-MEM plus 10% FCS was added to the cells. Twenty-four hours post transfection, the cells were used for performing DNA DSB and colony formation assays.

### 4.6. Total Protein Extraction, Western Blotting

Twenty-four hours after transfection with esiRNA (esiCTRL and esiSYNM), the cells were re-seeded in 3D lrECM. The next day, the cells were irradiated with 6-Gy X-rays or left unirradiated. The harvesting of total cell lysates was performed after 0.5, 1, 2, 6 or 24 h post irradiation. Whole cell lysates, SDS-PAGE and Western blotting were performed, as previously described in [11]. Original uncropped image of Western Blot can be found at Appendix A.

### 4.7. 3D Colony Formation Assay

The 3D colony formation assay was applied for measuring the ability of a single cell to form a colony, as published in [11]. The cells were transfected with esiRNA/siRNA (see esiRNA and siRNA transfection) and the next day were embedded into 0.5 mg/mL lrECM in 96-well plates. The cells were irradiated 24 h post reseeding and kept for several days (cell line-dependently) in 8.5% CO_2_ at 37 °C. Each point on the survival curve represents the mean surviving fraction from at least three independent experiments.

### 4.8. Foci Assay

For determination of DSBs, cells were stained for γH2AX and 53BP1, as described in [11]. Further details can be found in Appendix A.

### 4.9. Synenim Constructs and Stable Transfection

Human mCherry–Synemin was kindly provided by R. J. Bloch (University of Maryland, College Park, MD, USA). Different constructs were generated using PCR. The set of primers can be found in Appendix A. The primers used for the generation of all the constructs were designed with the HindIII restriction site for the forward primer and the BamHI restriction site for the reverse primer. Synemin putative phosphorylation sites were mutated using QuickChangeII XL Site directed Mutagenesis (#33790, Agilent, Santa Clara, CA, USA) by modifying one or two residues. The primers used for each mutation are listed in the Appendix A. The mutated sites were confirmed by sequencing. Stable transfection of the synemin constructs was performed as published in [47] using lipofectamine2000 (Invitrogen) and G418 (#A1720, Sigma-Aldrich, Taufkirchen, Germany) for the selection of cells.

### 4.10. DRGFP and pimEJ5GFP-Based Chromosomal Break Reporter Assay

For measuring HR and NHEJ activity, DRGFP and pimEJ5GFP plasmids were stably transfected, respectively, to generate an isogenic cell line pair (Cal33), as published in [50]. To measure HR/NHEJ-mediated repair, the cells were transiently transfected with pcDNA3BMyc-NLS-ISceI to express the I-SceI endonuclease for DSB induction [51]. Along with I-SceI, the pEGFP-N1 plasmid (Clontech, Mountain View, CA, USA), was transfected for determining transfection rates. Transfection was performed using lipofectamine2000. At 8 h after esiRNA transfection, the cells were transfected with I-SceI and pN1 plasmids. Four hours thereafter, the cells were trypsinized and embedded into 0.5 mg/mL lrECM. In the case of Synemin overexpression (mCherry–Synemin and mCherry–C1), both plasmids were transfected together with the pN1 and I-SceI. Transfection was performed using lipofectamine2000, according to the manufacturer’s protocol. The cells were then reseeded in 3D, trypzinized at 72 h and subjected to flow cytometry (FACS Canto; BD Biosciences, San Jose, CA, USA). Per sample, 2 × 10^4^ events were measured. GFP-positive cells were normalized to pEGFP-N1-positive cells and analysis was performed using FlowJo software (version 7.6.2; BD, San Jose, CA, USA).

### 4.11. Proximity Ligation Assay (PLA)

PLA was performed according to the manufacturer’s protocol using the Duolink^®^ PLA protein detection kit with PLA PLUS and MINUS Probes for mouse and rabbit (DUO92101-1KT, Sigma-Aldrich). More details can be found in Appendix A.

### 4.12. Immunoprecipitation

For the precipitation of the antibody–protein complex out of a cell lysate, protein A/G-beads (#PRAG25-A5-5, Alpha Diagnostics Intl., San Antonio, TX, USA) were used, as described in [48]. Further details can be found in Appendix A.

### 4.13. Cell Cycle Analysis

Synemin-depleted un- and irradiated SAS cells cultured for 24, 48 and 72 h were incubated with 10 mM BrdU (BD Biosciences) for 10 min. Cells were harvested using 1× trypsin/EDTA, fixed in ice cold 80% ethanol for 10 min and incubated for 10 min with 0.01% RNase A (Sigma-Aldrich), followed by 30 min incubation with 2 N HCl (Sigma-Aldrich) and 0.5% Triton-X-100/PBS (Roth, Karlsruhe, Germany). Subsequently, mouse anti-BrdU antibodies and propidium iodide (Sigma-Aldrich) were added for the detection of incorporated BrdU and total DNA content. Cell cycle distribution was determined using an FACS Calibur (BD Biosciences) and analyzed using the FlowJo software (version 7.6.2).

### 4.14. Data Analysis

The means ± standard deviation (SD) of at least three independent biological experiments (indicated as *n*) were calculated with reference to controls defined in total numbers or 1.0. For statistical significance, two-sided Student’s *t*-test was performed using Microsoft Excel 2003. A *p*-value of less than 0.05 was considered statistically significant. For foci experiments, one-way ANOVA, followed by post-hoc analysis using Turkey’s correction, was used.

## 5. Conclusions

In summary, our data pinpoint the multifaceted roles of intermediate filaments and the regulatory complexity of DNA repair processes. ATM-dependently, synemin serves as a scaffold for DNA-PKcs and co-determines DNA double strand break repair upon genotoxic injury. The interactions of synemin, DNA-PKcs and ATM further underscore the concept of cytoarchitectural elements as key co-regulators of nuclear events. In this context, future investigations are warranted to elucidate the associated aspects of the DNA damage response, such as the cytoplasmic-to-nuclear shuttling of DNA repair proteins and chromatin reorganization.

## Figures and Tables

**Figure 1 cancers-12-01717-f001:**
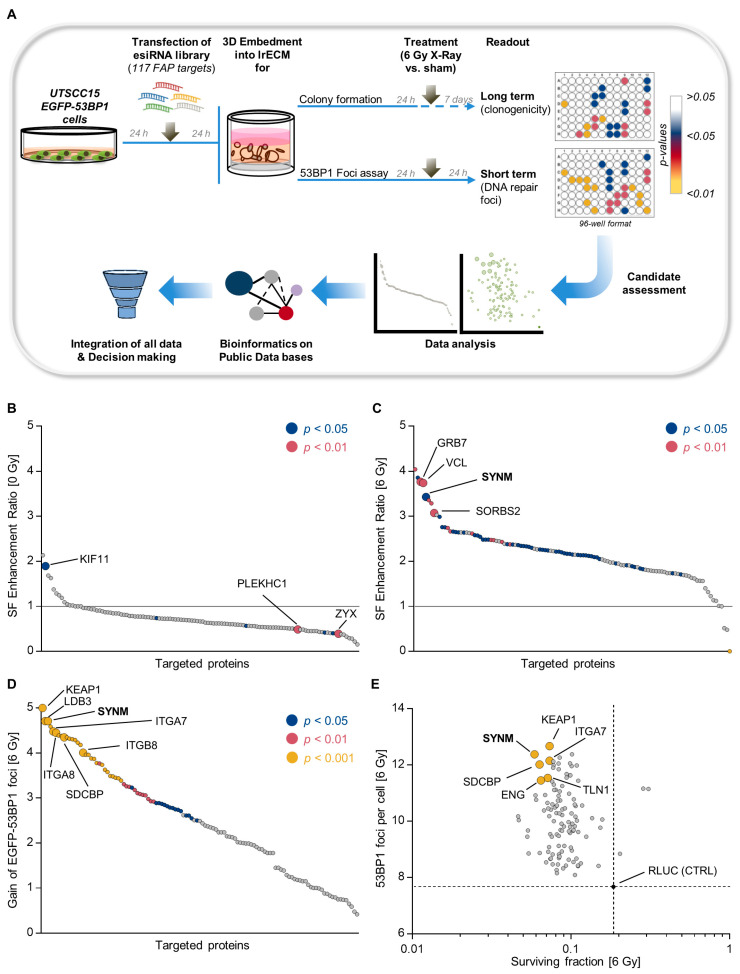
Identification of focal adhesion proteins affecting cell survival, radiosensitivity and DNA repair. (**A**) Workflow of 3D high-throughput RNAi screening (3D HTP-RNAi-S); (**B**) Surviving fraction-related enhancement ratios of EGFP-53BP1-expressing UTSCC15 cell cultures in response to focal adhesion protein (FAP) knockdowns (*n* = 4), the data and *p*-values for which are provided in Appendix A; (**C**) Surviving fraction-related enhancement ratios of FAP knockdown cell cultures exposed to 6-Gy X-rays (*n* = 4), the data and *p*-values for which are provided in Appendix A; (**D**) Gain of residual 53BP1 foci number per cell (24 h after irradiation) in FAP knockdown cell cultures irradiated with 6-Gy X-rays (53BP1 foci in controls were subtracted from the total number of foci) (*n* = 4); (**E**) Scatter plot displaying the relation between 53BP1 residual foci/cell and surviving fraction upon FAP knockdown and 6-Gy irradiation. Main selected candidates with high foci/cell and low cell survival are indicated. SF, surviving fraction; CTRL, control.

**Figure 2 cancers-12-01717-f002:**
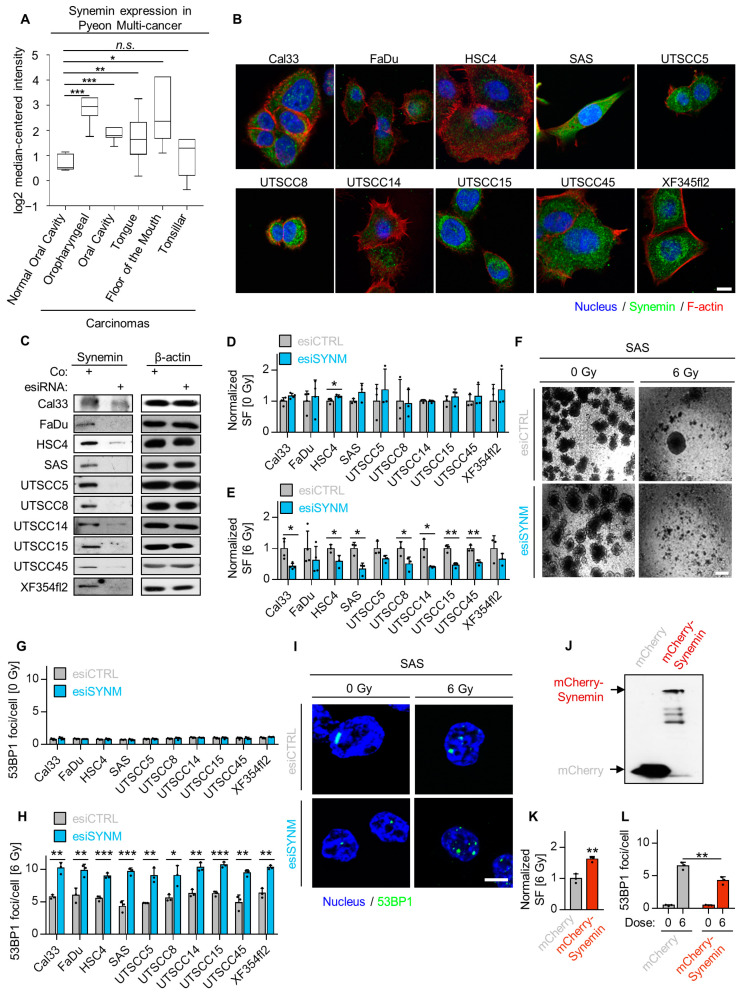
Synemin essentially controls radiosensitivity and DSB repair. (**A**) Analysis of synemin mRNA expression in head and neck carcinomas in comparison to corresponding normal tissue using Oncomine database; (**B**) Immunofluorescence staining of synemin distribution (green) in a panel of HNSCC cell lines. Cells were counterstained with Phalloidin (F-actin, red) and DAPI (nucleus, blue) (bar, 20 µm); (**C**) Immunoblots with knockdown efficiencies in a panel of HNSCC cell lines; (**D**) Normalized plating efficiency of a panel of HNSCC cell lines upon synemin inhibition (*n* ≥ 3); (**E**) Colony formation ability of 6-Gy X-ray irradiated 3D lrECM HNSCC cell cultures after esiRNA-mediated synemin depletion; (**F**) Representative phase contrast images of 3D lrECM SAS cell cultures (bar, 500 µm); (**G**) Spontaneous foci per cell in a panel of HNSCC cell lines upon synemin inhibition (*n* = 3); (**H**) Effect of synemin silencing on residual 53BP1 foci (24 h after irradiation) in a panel of 6-Gy irradiated 3D lrECM HNSCC cell lines; (**I**) Representative immunofluorescence images of residual 53BP1 foci (bar, 10 µm); (**J**) Immunoblot of mCherry–Synemin and mCherry empty vector expression; (**K**) Colony formation ability of SAS mCherry–Synemin transfectants, relative to SAS mCherry controls (6-Gy X-rays); (**L**) Residual 53BP1 foci (24 h after irradiation) in SAS mCherry–Synemin transfectants exposed to 6-Gy X-rays. Data are presented as mean ± SD (*n* = 3; two-sided *t*-test; * *p* < 0.05, ** *p* < 0.01, *** *p* < 0.001).

**Figure 3 cancers-12-01717-f003:**
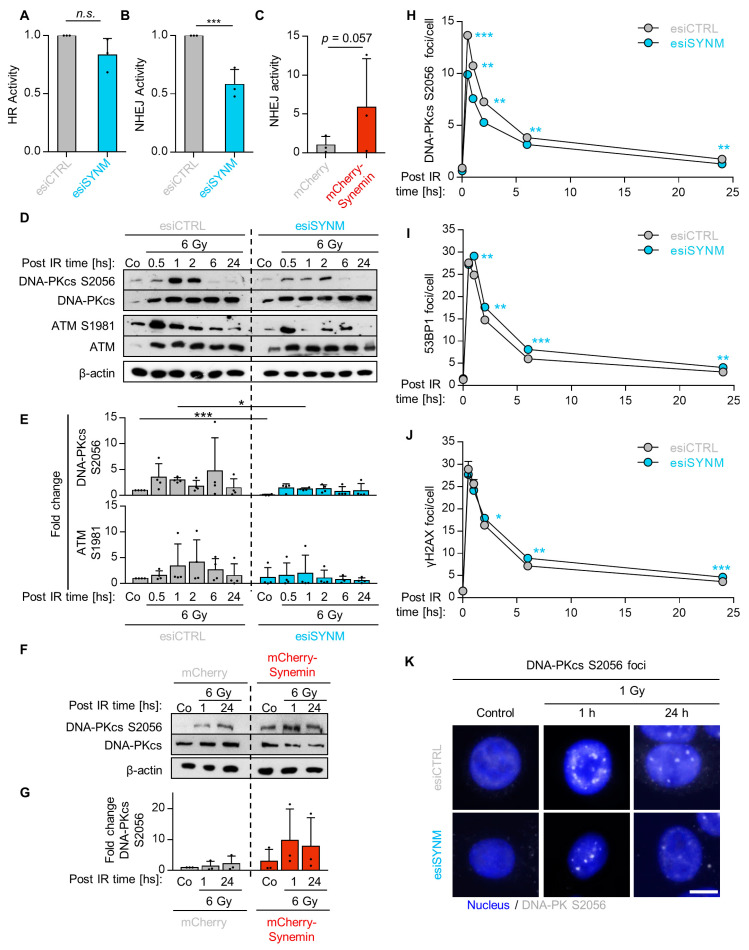
Synemin functions in non-homologous end joining. GFP-based reporter assays for (**A**) HR and (**B**) NHEJ. Cal33 cells stably transfected with DRGFP or pimEJ5GFP recombinant plasmids were depleted of synemin. The number of GFP-positive cells was analyzed by FACS, provided in Appendix A; (**C**) NHEJ activity in mCherry–Synemin-overexpressing Cal33-pEJ5GFP cells. Analysis performed by FACS as indicated under (**A**,**B**); (**D**,**E**) Immunoblots and fold changes from synemin-depleted and 6-Gy irradiated SAS cells showing total and/or phosphorylated forms of DNA-PKcs, ATM. β-actin served as loading control; (**F**,**G**) Immunoblot and fold change of DNA-PKcs from whole cell lysates of 6-Gy X-ray irradiated and mock-treated SAS mCherry–Synemin transfectants. β-actin served as loading control; (**H**–**J**) Kinetics of DNA-PKcs S2056, 53BP1 and γH2AX foci upon synemin knockdown at different time-points post 1-Gy X-rays in SAS cells; (**K**) Representative immunofluorescence images of residual DNA-PKcs S2056 foci of synemin knockdown and control cell cultures 1 h after 1-Gy X-rays (bar, 10 µm). Data are presented as mean ± SD (*n* = 3; two-sided *t*-test; ** *p* < 0.01, *** *p* < 0.001; n.s., not significant (*p* ≥ 0.05)).

**Figure 4 cancers-12-01717-f004:**
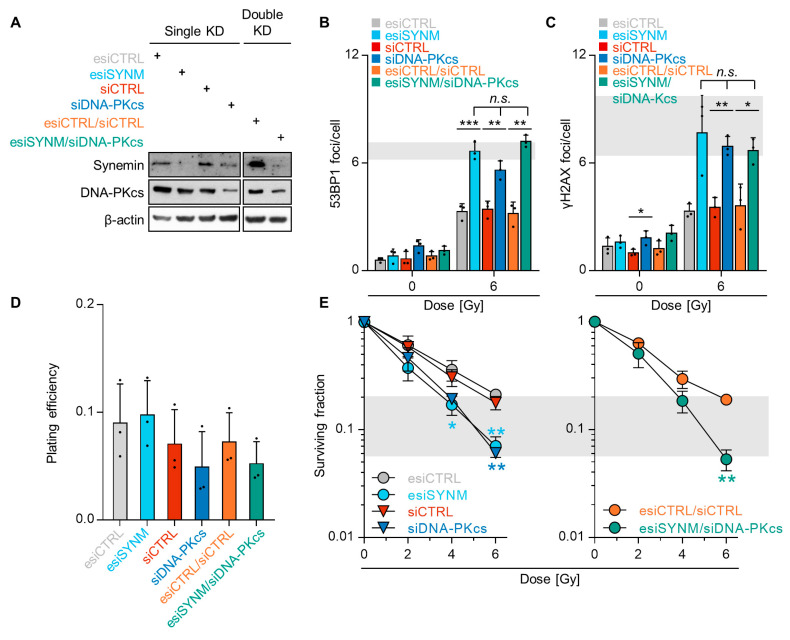
Synemin/DNA-PKcs co-control DSB repair and radiation survival. (**A**) Knockdown efficiencies of single and double esi/siRNA transfections of SAS cells. β-actin served as loading control; (**B**,**C**) Residual 53BP1 and γH2AX foci per cell (24 h after irradiation) upon single and double knockdown of synemin and DNA-PKcs in 6-Gy X-ray irradiated SAS cells. Transfection with single or double non-specific siRNA were used as controls; (**D**) Plating efficiency of SAS cells upon single and double knockdown of synemin and DNA-PKcs; (**E**) 3D colony formation ability upon single and double silencing of synemin and DNA-PKcs. Data are represented as mean ± SD (*n* = 3; two-sided *t*-test; * *p* < 0.05, ** *p* < 0.01, *** *p* < 0.001; n.s., not significant (*p* ≥ 0.05)).

**Figure 5 cancers-12-01717-f005:**
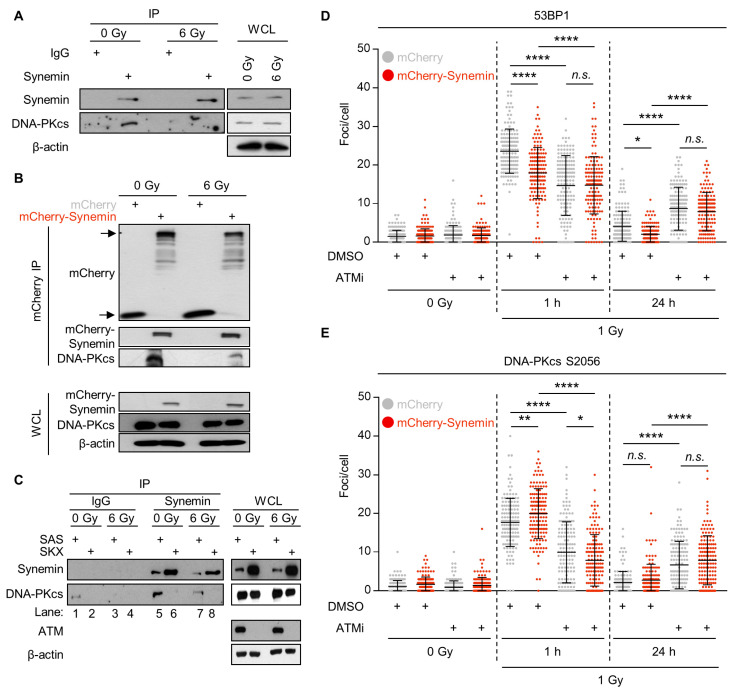
DNA-PKcs directly interacts with synemin and depends on ATM activity. (**A**) Immunoprecipitation (IP) of Synemin 1 h post 6-Gy X-rays. Western blot analysis showing expression of Synemin and DNA-PKcs in immunoprecipiates and whole cell lysates (WCL) from SAS cells; (**B**) Western blot on mCherry immunoprecipitates from 6-Gy irradiated mCherry-SAS and mCherry–Synemin-SAS cells at 1 h post irradiation. β-actin served as loading control; (**C**) Western blots on synemin immunoprecipitates from SAS and SKX cells 1 h after 6-Gy X-ray exposure; (**D**,**E**) 53BP1 and DNA-PKcs S2056 foci upon ATMi treatment (DMSO used as control) at different time points post 1-Gy X-rays in SAS mCherry and mCherry–Synemin transfectants. Data are represented as mean ± SD (*n* = 3; One-way ANOVA followed by post hoc test (Tukey multiple comparisons); * *p* < 0.05, ** *p* < 0.01, *** *p* < 0.001, **** *p* < 0.0001; n.s., not significant (*p* ≥ 0.05)).

**Figure 6 cancers-12-01717-f006:**
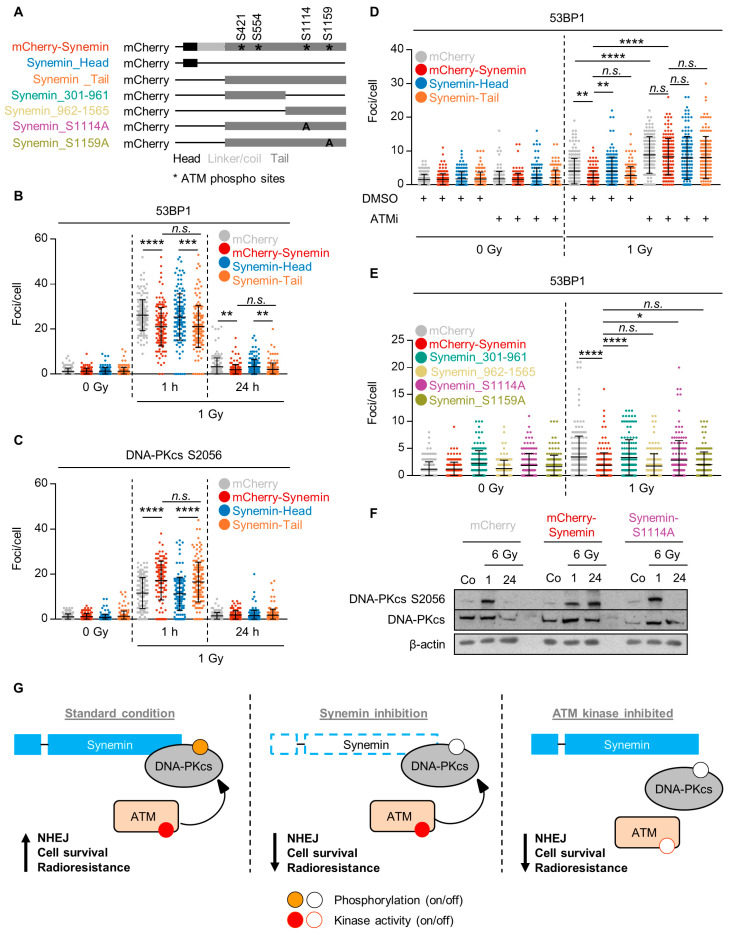
Synemin-mediated DNA repair depends on the S1114 phosphorylation site of synemin. (**A**) Design of different synemin constructs; (**B**,**C**) 53BP1 and DNA-PKcs S2056 foci kinetics in 1-Gy X-ray-irradiated SAS cells expressing mCherry–Synemin wildtype, mCherry–Synemin_Head or mCherry–Synemin_Tail (mCherry was used as control); (**D**) Residual 53BP1 foci (24 h after irradiation) in SAS cells expressing mCherry–Synemin wildtype, mCherry–Synemin_Head or mCherry–Synemin_Tail (mCherry was used as control) treated with ATMi and 1-Gy X-rays; (**E**) Residual 53BP1 foci (24 h after irradiation) in 1-Gy X-ray-irradiated SAS transfectants expressing mCherry–Synemin wildtype, mCherry–Synemin_301–961, mCherry–Synemin_962–1565, mCherry–Synemin_S1114A and mCherry–Synemin_S1159A (mCherry was used as control); (**F**) Western blotting of lysates from 6-Gy irradiated (1 and 24 h) and unirradiated SAS cells expressing mCherry, synemin-wt and synemin-S1114A; (**G**) Schematic depiction of how synemin interacts with DNA-PKcs and ATM for controlling NHEJ, cell survival and radioresistance in HNSCC cells, and that either synemin or ATM targeting renders cells equally radiosensitive. Results are presented as mean ± SD (*n* = 3; One-way ANOVA followed by post hoc test (Tukey multiple comparisons); * *p* < 0.05, ** *p* < 0.01, *** *p* < 0.001, **** *p* < 0.0001; n.s., not significant (*p* ≥ 0.05)).

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
