# Peer review of "The Intermediate Filament Synemin Regulates Non-Homologous End Joining in an ATM-Dependent Manner"

_cancers, 2020, doi:10.3390/cancers12071717_

Round 1

Reviewer 1 Report

The authors have submitted a well designed and clearly presented series of experiments which stem from a 3D based radiosensitivity screen of focal adhesion proteins (with two independent endpoints)

They identified synemin from this screen and demonstrated that suppression and over expression of this protein altered both cellular radiosensitivity and formation of 53BP1 and gH2AX foci. These effects are demonstrated in a suitably broad panel of HNSCC tumour cell lines.

The subsequent demonstration that this effect appears to be mediated via NHEJ and has an epistatic effect with DNA-PK is clear and robust.

My main comments are:

There are some minor grammatical corrections that could be made throughout the document

The authors argue that a particular strength of this work is that 3D based studies have advantages over 2D based experiments. It would therefore be interesting to see the effects of synemin suppression in a 2D model - ie would a 2D based screen have failed to identify this target?

It would be helpful to have more detail as to why this particular gene was selected from the screen rather than some of the other high ranking genes?

Formal statistical correlation and comment on the relationship between the 53BP1 and survival screening endpoints. 

The relative increase in gH2AX foci formation seems to be quite modest compared with the 40% reduction in NHEJ - can the authors explain/comment on this please?

There appears to be quite a big difference in the magnitude of sensitisation seen with SYNM knockdown alone in Fig 4E compared 2E. Is this correct, and if so why might this be?

Reviewer 2 Report

The manuscript from Deville and colleagues reports a comprehensive study of the role of focal adhesion proteins in the execution of the DNA damage response to ionizing radiation. An RNAi screen of 117 targets identified synemin, among others, that was then studied in detail. The authors provide robust data in 10 HNSC cell lines showing that silencing synemin increases radiosensitivity measured by two independent assays. They perrom mechanistic studies to follow up a putative interaction of synemin with DNA-PKcs, which they confirm and demonstrate is necessary for DNA PKcs phosphorylation by ATM required for action in NHEJ. The studies are well -described, well-documented, and provide substantial evidence that this novel biology is important in the execution of NHEJ independent of cell cycle effects. All the appropriate controls and statistical analysis are reported. I have only a few questions that could clarify the domain of regulation.

In Figure 2B synemin is localized in a panel of cells using immunofluorescence, showing predominantly cytoplasmic localization as expected, but there is no comment or discussion of whether the interaction with DNA PKcs occurs in the cytoplasm or the nucleus--if the latter it would appear that synemin compartmentalization is important. I would expect a nuclear fractionation would be useful to address this and perhaps high magnification confocal microscopy.  This would be particularly compelling if there is co-localization with DNA damage foci, like the 53BP1.

Second, the authors make a statement that there a functional context /relationship for the RNAi screen between colony formation and 53BP1 foci represented in Fig 1 E, but I don't see it and there is no correlation analysis. I would suggest representing these data only for the significant hits to test the presumably positive relationship using a Pearson or Spearmen correlation. It is a minor point though, if that analysis does not substantiate the claim, I would suggest deleting the comment.

Last, I would like the authors to use the term 'colony assay' rather than clonogenic survival because the cells are plated 24 hr before irradiation, greatly increasing the likelihood of at least doublets present at irradiation. Moreover cells in matrigel do move and aggregation can contribute to spheres that are counted.

Reviewer 3 Report

The authors of this manuscript propose a novel role for the intermediate filament protein synemin, in regulating DNA double strand break repair by non-homologous end-joining (NHEJ) in head and neck squamous cell carcinoma (HNSCC) cells. This was discovered using an RNAi screening approach targeted against 117 focal adhesion proteins, and then monitoring both clonogenic survival and 53BP1 foci formation. Screening results were then validated in 10 HNSCC cell lines, which demonstrated that synemin was required for maintaining cell survival following x-ray irradiation, and led to increased amounts of 53BP1 foci. The authors go on to show that synemin is required for auto-phosphorylation of DNA-Pk via a direct interaction and in an ATM-dependent manner, that this is required for a timely recruitment of 53BP1 to DNA double strand break foci, and this ultimately promotes cell survival.

In general, the manuscript is well written, and describes an interesting finding linking the intermediate filmament synemin in regulating the cellular DNA damage response. However, there are some major limitations that need to be addressed, such as the limited time points and radiation doses that are utilised, the different cell lines that are used for testing certain mechanistic aspects, and the correlation of synemin in controlling DNA-Pk phosphorylation with NHEJ capacity. These are detailed more below.

Major comments:

  1. There are a number of questions related to the RNAi screen presented in Figure 1. It is unclear why the authors chose to use UTSCC15 cells overexpressing 53BP1 (apart from ease of foci analysis) in their screen, but particularly the level of overexpression of the protein relative to endogenous 53BP1. This should at least be shown by immunoblotting analysis. The authors also use a relatively high dose of x-ray irradiation (6 Gy) in their RNAi screen (Figure 1), which arguably is not clinically relevant. Nevertheless, it is not clear in the figure legend whether the number of replicates stated (n=4), relates to those conducted within the same experiment, or are as separate biological experiments. These points should be clarified.
  2. The authors in Figures 2C-I validate the impact of synemin depletion on reducing clonogenic cell survival and causing an elevation in 53BP1 (presumably, this is 24 h post-irradiation which needs to be clearly stated in the text and/or legend?) following 6 Gy X-rays in 10 HNSCC cell lines. The number of cell lines utilised is fully acknowledged however, analysing the responses at a single dose (survival) and time point (53BP1) is very restrictive. It is recommended that at least two or three of these cell lines (particularly Cal33 and SAS that are largely employed later in the manuscript) are analysed in more detail using a full radiation dose response by clonogenic assays, and kinetics of 53BP1 foci formation determined over a time course post-irradiation. The appropriate statistical analysis should then be performed on this collective data.
  3. In Figure 3, the authors analyse the impact of synemin depletion on NHEJ capacity, ATM/DNA-Pk phosphorylation and DNA repair foci formation, although there are a number of unanswered questions about this. Firstly, the alteration in NHEJ is markedly affected by both synemin depletion and overexpression (Figure 3A), but this doesn’t appear to correlate well with impact on DNA-Pk phosphorylation and DNA-Pk/53BP1 foci formation. In fact, 53BP1 recruitment appears slightly delayed and only marginally higher in synemin depleted cells (Figure 3I), and a relatively minor impact on gH2AX formation is observed (Figure 3J). This does not appear to have a strong correlation with the ~40 % reduction in overall NHEJ capacity. Do the authors have any explanation for this, or have they examined the downstream events (e.g. ligase IV activity?) which may be impacted on synemin loss? In Figure 3D, there appears to be a dramatic increase in the levels/stabilisation of DNA-Pk in cells without synemin in the absence of radiation (compared to si-control cells), but then the protein is very unstable at 6-24 h post-irradiation. These data are not highlighted or discussed, particularly as to whether synemin actually controls DNA-Pk stability and/or degradation. Additionally, the kinetics of phosphorylation of DNA-Pk by immunoblotting in si-control cells (Figure 3D) where this peaks at 1 h post-irradiation and reduces to zero from 6-24 h, do not appear to be reflected in the quantified fold change (Figure 3E). Here, DNA-Pk phosphorylation actually peaks at 0.5 h post-irradiation, decreases slightly at 1-2 h, but then increases again at 6 h. The same comment can be made of ATM phosphorylation, where the immunoblotting does not match up well at all with fold quantification. This contradictory data needs resolving (e.g. by incorporation of an immunoblot that actually represents the quantified data). It is nevertheless recommended (also will be pointed out in Comment 5) that these critical data be reproduced in another cell line (e.g. SAS cells).
  4. Similar to Comment 2, there is a limited analysis of the impact of synemin and DNA-Pk depletion on 53BP1/H2AX foci formation in SAS cells in Figures 4B-C (only 24 h post-irradiation?), and the impact of expression of synemin truncation mutants on foci formation at limited time points, particularly in Figures 6D-E. It is advised that these are expanded to include at least one or two additional time points post-irradiation to get a full appreciation of the impact of the gene manipulation on DNA repair protein recruitment.
  5. As stated in my summary above, the subsequent figures in the manuscript appear to pick and chose which cell line is used to test the mechanism of synemin in promoting NHEJ, without any specific validation. Specifically, Figure 3 uses Cal33 cells to analyse NHEJ activity and DNA-Pk phosphorylation, Figure 4 analyses survival and 53BP1/H2AX foci formation in SAS cells, and Figures 5/6 utilise synemin overexpressing SAS cells to analyse 53BP1/DNA-Pk foci formation. For consistency, it would make sense to utilise the same cells throughout. Therefore as a suggestion, the data in Figure 3 (particularly the impact of synemin depletion on kinetics of DNA-Pk phosphorylation by immunoblotting and foci analysis, which is key to the overall hypothesis) should at least be reproduced in SAS cells.
  6. It is noticeable in Figures 5D-E and 6B-E that the dose of radiation used to analyse 53BP1/DNA-Pk foci formation is reduced to 1 Gy. This is not consistent with the 6 Gy dose largely used throughout the manuscript, but also within the same figures. The authors should again provide some consistency and use the same dose, or provide a compelling argument why it was necessary to reduce the dose.
  7. One critical factor which appears neglected throughout, is the obvious difference in cellular localisation of synemin (cytoskeletal) and DNA-Pk (predominantly nuclear). This is reflected to some extent in Figure 2B, although there does appear to be a small proportion of synemin present in the nucleus of some cells (e.g. HSC4 and XF354fl2). Whilst the authors demonstrate that synemin and DNA-Pk appear to interact (Figure 5A-C), there is no indication of the relative proportions of the interacting proteins. Additionally, the cell extract levels of the proteins are on separate figures/blots, so these are not directly comparable. The authors should address this issue. As an extension of this point, have the authors examined potential co-localisation of synemin and phosphorylated DNA-Pk pre- and post-irradiation by immunofluorescence? Minimally, it is advisable that biochemical fractionation of cell extracts should be employed and localisation of synemin and (phosphorylated) DNA-Pk analysed within cytoplasmic and nuclear fractions pre- and post-irradiation using quantitative immunoblotting.

Minor comments:

  1. The immunoblots in Figure 4A for synemin are of poor quality, and should be replaced. The knockdown of DNA-Pk also appears not particularly efficient, but at least should be quantified and efficiency stated.
  2. The immunoblots demonstrating interaction of synemin and DNA-Pk in Figure 5A, should looked to be improved.
  3. The authors should be clear in the Figure legends the specific cell lines that are used for each experiment, as well as the time points analysed post-irradiation, where necessary.

Round 2

Reviewer 3 Report

I appreciate the attempts the authors have made to revise their manuscript in the relatively short space of time, however there are still some remaining major concerns that in my opinion have not been adequately addressed. A number of additional experiments need to be included for a more thorough comprehensive analysis, and I would advise that the authors spend the required amount of time to complete these.

Major comments remaining (directly related to points in my previous review):

  1. The authors need to be more transparent about numbers of replicates. I would not consider replicates within the same experiment (2 wells per condition in Figure 1), to be independent biological experiments. The authors should therefore at least clearly state the composition of the replicates in the manuscript, as they have done in their response (n=2, 2 wells per condition).
  2. For Figure 2, I can not see that the legend has been appropriately altered to clearly state the 24 h exposure. I also still recommend (as previously requested) that clonogenic survival dose response and γH2AX/53BP1 foci formation be determined over a full time course post-irradiation in multiple cell lines (SAS, Cal33 particularly), rather than analyses presented using a single cell line/end-point, which would clearly validate the results of the screen.
  3. Regarding levels/stabilisation of DNA-Pk, I appreciate that the densitometries have been added in the Supplementary (as Figure S8). However, and as pointed out in my initial review, these do not correlate at all well with Figure 3D where the protein appears very unstable at 6-24 h post-irradiation in the absence of synemin, in contrast to the quantification that shows these actually increase? The same comment remains relating to quantification of phosphorylated ATM and DNA-Pk protein levels. This brings into question the significant variability in the data, and their overall meaning. Additionally, no attempt has be made to reproduce these data in another cell line (e.g. SAS cells), as initially requested.
  4. Regarding Figures 4B-C, whilst I appreciate the authors suggest that these are residual foci, the data are still very limited in scope. The impact of synemin and DNA-Pk depletion on 53BP1/γH2AX foci formation in SAS cells should be expanded to include additional time points post-irradiation to understand whether the foci are eventually resolved, or are very persistent, to get a full appreciation of gene manipulation on NHEJ capacity.
  5. Regarding my comment on choice of cell lines, I do indeed appreciate the authors experiments in 3D cell cultures. My intention, as in comment 2, was to point out that specific biological end-points (survival following a radiation dose response; kinetics of γH2AX/53BP1/DNA-Pk foci formation) be reproduced in multiple cell lines to ensure that this is not a cell-line specific effect. This therefore, has still not been adequately addressed in the revised manuscript.
  6. I accept the authors argument of using a 1 Gy X-ray dose to simplify quantification, but recommend that this is clearly stated in the main text.
  7. I appreciate the inclusion of the new data in Figures S6C-D and S10C-10 which further strengthen that synemin and DNA-Pk interact. There is still no information included though on Figures 5A-C, to understand the relative proportions (%) of the interacting proteins as quantified through biochemical analysis. The authors should address this issue, and as previously suggested, biochemical fractionation of cells to examine localisation of synemin and (phosphorylated) DNA-Pk analysed within cytoplasmic and nuclear fractions pre- and post-irradiation using quantitative immunoblotting should be performed.

Regarding minor comments, and unless I am mistaken, immunoblots for Figure 4A (synemin) and Figure 5A (synemin and DNA-Pk) have not changed and alternatives are still suggested to replace these.
